# Priming conditions shape breadth of neutralizing antibody responses to sarbecoviruses

Janice Zhirong Jia[1,12], Chee Wah Tan [2,12], Samuel M. S. Cheng[3], Haogao Gu [3], Aileen Ying Yan Yeoh[2], Chris Ka Pun Mok [4,5], Yanqun Wang[6], Jincun Zhao [6], Nancy H. L. Leung [7], Benjamin J. Cowling [7], Leo L. M. Poon [1,3,8], David S. C. Hui [9,10], Linfa Wang [2,13], Malik Peiris [3,8,13] & Sophie A. Valkenburg [1,11,13] ✉

Vaccines that are broadly cross-protective against current and future SARS-CoV-2 variants of concern (VoC) or across the sarbecoviruses subgenus remain a priority for public health. Virus neutralization is the best available correlate of protection. To define the magnitude and breadth of cross-neutralization in individuals with different exposure to SARS-CoV-2 infection and vaccination, we here use a multiplex surrogate neutralization assay based on virus spike receptor binding domains of multiple SARS-CoV-2 VoC, as well as related bat and pangolin viruses. We include sera from cohorts of individuals vaccinated with two or three doses of mRNA (BNT162b2) or inactivated SARS-CoV-2 (Coronavac or Sinopharm) vaccines with or without a history of previous SARS-CoV-2 or SARS-CoV-1 infection. SARS-CoV-2 or SARS-CoV-1 infection followed by BNT162b2 vaccine, Omicron BA.2 breakthrough infection following BNT162b2 vaccine or a third dose of BNT162b2 following two doses of BNT162b2 or Coronavac elicit the highest and broadest neutralization across VoCs. For both breadth and magnitude of neutralization across all sarbecoviruses, those infected with SARS-CoV-1 immunized with BNT162b2 outperform all other combinations of infection and/or vaccination. These data may inform vaccine design strategies for generating broadly neutralizing antibodies to SARS-CoV-2 variants or across the sarbecovirus subgenus.

Vaccine mediated protection against COVID-19 is primarily determined by neutralizing antibody titer[1]. Neutralizing antibodies directly interfere with the interaction of SARS-CoV-2 virion Spike (S) receptor binding domain (RBD) with the Angiotensin Converting Enzyme 2 (ACE2) receptor, which is predominantly expressed in the lung, gut and heart. However, VoC have emerged that have accumulated mutations in the S, especially the RBD, which results in escape from neutralizing antibodies generated by the ancestral SARS-CoV-2 virus from Wuhan in 2019. The greatest threat so far has been VoC Omicron (B.1.1.529) and its subvariants. It was first reported in November 2021,

and contains about 50 non-silent mutations and over 2/3 of these mutations are in the S domain[2]. In 2002, SARS-CoV (herein called SARS-CoV-1) emerged and caused 8000 infections with public health measures abating the outbreak[3]. In contrast, SARS-CoV-2, has infected over 500 million people within 24 months, and the severity of the pandemic countered by over 12 billion doses of highly effective COVID-19 vaccines. The most predominantly used SARS-CoV-2 vaccines, are inactivated whole virion adjuvanted vaccines (e.g., Coronavac), which have been widely administered due to ease, scalability and lower cost of production and relative thermostability. The second most widely

used vaccines are Spike encoding mRNA lipoparticle vaccines[4]. However, there is at least 10-fold difference in neutralizing antibody titers elicited by these two types of vaccines[5,6], which results in lower protective efficacy of inactivated vaccines[1]. This lower protection may be due to mismatch between vaccine induced antibodies and native conformation of the virus S, as β-propiolactone which is used to split the SARS-CoV-2 virion for inactivation may affect S conformation. In contrast, mRNA vaccines are expressed by host cells in a S-2P stabilized conformation as a pre-fusion form directly representing the S as it appears on virions.

S-specific antibodies wane, faster in the first few months and slower later, after infection[7]. This is expected as plasmablast responses contract and a stable memory B cell pool forms with reduced antibody output during late convalescence. However, antibody waning post-vaccination, across different vaccine formats is also substantial, leading to reduced protective efficacy against infection at 6 months post 2-dose vaccination. Reassuringly protection against severe disease is not as compromised and other immune mechanisms such as T cells and non-neutralising antibody functions may contribute to this longer duration of protection against severe disease[8–10]. To maintain protection against mild disease, booster third-dose vaccinations were recommended in mid-2021 in Israel, which led to an 11x increase in protection within 2 weeks post vaccination[11], and have since become required for full vaccination status in many developed countries. However, waning antibody responses have again occurred, and fourth dose vaccination is now (in mid-2022) being considered for at risk individuals in some countries. We therefore must define the immune priming-boosting strategies that elicit broadly reactive antibodies to SARS-CoV-2 to counter future variants and inform next generation vaccines that may protect us from other sarbecoviruses.

The phenomenon of Original Antigenic Sin (OAS)[12] occurs where the antigen we are first primed against limits our capacity to respond to novel antigenic epitopes presented by closely related variants, due to clonal competition of B cells at subsequent encounters. OAS can play a significant role in capping influenza vaccine efficacy[13], whereby antigenic focusing can occur with repeated vaccination[14]. The impact of OAS is impacted by antigenic distance[13], the serological 'space' a virus may occupy, and re-vaccination with a distinct serotype may then justified. This forms the basis of strain updates to seasonal influenza vaccines. Heterologous vaccine regimes of alternating formats and adjuvants can improve vaccine responses by recruiting existing memory and generating new responses leading to synergistic results[15,16]. Coronavac includes an Alum adjuvant which acts as a TLR7 agonist to improve antigen presentation[17]. The majority of adults are seropositive to related beta-"common cold" coronaviruses (CCoV), OC43 or HKU-1, and S-specific CCoV antibodies are boosted by SARS-CoV-2 infection and are therefore cross-reactive to some extent[18], but they do not mediate a protective response to reduce the duration of illness nor viral shedding[19,20]. The observation that mRNA vaccination with COVID-19 vaccines of SARS-CoV-1 convalescent individuals led to generation of broadly neutralizing antibodies[21], has provided hope for pan-sarbecovirus vaccines as the "holy grail" for next generation vaccines. In this study, we sought to identify immune priming conditions that generate broadly neutralizing antibody responses.

## Results

### Inhibition of ACE2 binding to ancestral, VoC Beta and Delta RBD by antibody in the multiplex sVNT assay correlates with the plate sVNT assay

Plaque reduction neutralisation (PRNT) assays are a gold standard to assess antibody activity against blocking virus entry, however due to technical limitations the PRNT assay can be difficult to perform against a panel of viruses that have different replicative fitness and host range/cell lines. Furthermore, some sarbecoviruses have been sequenced but infectious virus not isolated (e.g., Bat CoV RaTG13)[22]. We therefore used a surrogate virus neutralization test (sVNT) in a multiplex format, which uses recombinant protein of receptor binding domain (RBD) of different SARS family viruses to assess antibody inhibition of ACE2 binding.

To assess the relative affinity of different RBD proteins for the human ACE2 (Fig. 1a), a 16-plex panel of RBD proteins representing SARS-CoV-2, related variants of concern (VoC), clade 2 bat and pangolin derived viruses, in addition to the SARS-CoV-1 virus and related clade 1 bat viruses (Bat CoV WIV-1, RsSHC014, LYRa11, Rs2018B), were tested for neutralizing activity by immune plasma from different priming conditions (Table 1, Supplementary Data 1). The protein homology of the RBD panel (Table 2)[23] ranges from 1 to 3 amino acid (aa) differences for a 99.6 to 98.7% conservation for VoCs Alpha to Mu, whilst Omicron BA.1 has 15 aa differences and is only 93.3% conserved versus SARS-CoV-2. Whilst clade 1 viruses have 55 to 60 aa differences in the RBD and are 73.1 to 75.3% conserved versus SARS-CoV-2.

Plasma samples from individuals convalescent from mild ancestral SARS-CoV-2 infections elicited neutralising antibody responses to SARS-CoV-2 and related VoC Alpha, Delta and Lambda, but minimal responses to other SARS-CoV-2 VoCs, bat sarbecoviruses or SARS-CoV-1 (Fig. 1a, b). The neutralization response was relatively short-lived, from 30–60 days post infection, with most antibody responses below 20% inhibition by day 80–270.

Authentic SARS-CoV-2 virus based PRNT assays were assessed versus the plate and bead based sVNT assays, which have previously shown to account for more than 90% of total neutralising antibodies[24,25] across different immune cohorts[23,26]. Live virus neutralisation strongly correlated with the plate format[27] ($r = 0.89$) and bead format ($r = 0.9$) sVNT assays (Fig. 1c, d). The plate and bead sVNT assays were also well correlated for the ancestral virus (Spearman correlation $r = 0.85$, Fig. 1e), VoC Beta ($r = 0.83$) and Delta ($r = 0.76$) (Supplementary figure 1a, b). However, Bland Altman analysis for assay comparability between the plate format and bead format sVNT assays (Fig. 1f), showed that the plate sVNT measures 12.53 units more than the bead sVNT. Although the biases between two assays are non-zero and non-linear by Bland Altman, the Spearman correlation (rank-based) is significant between them (Fig. 1e), suggesting the agreement between two methods by rank are good.

However, these correlations did not extend to the Omicron BA.1 bead sVNT assay which had weak correlations with Omicron BA.1 plate assay ($r = 0.18$) or PRNT ($r = 0.51$) assays (Supplementary fig. 1a, b), whilst the plate format of the sVNT Omicron assay correlated with PRNT results ($r = 0.71$, Supplementary fig. 1a, b). Therefore, hereafter we used the RBD of BA.1 Omicron in a fixed plate based commercial assay, whilst other RBDs were assessed in parallel in the multiplex bead format for further analysis.

### mRNA vaccination increases antibody breadth dependent on priming conditions

The breadth of antibody responses from alum adjuvanted inactivated whole virion vaccine, Coronavac (from SinoVac) was compared to the mRNA Spike lipoprotein vaccine BNT162b2 in previously infection naïve individuals (Fig. 2a). BNT162b2 vaccination (Fig. 2a) significantly boosted neutralizing antibodies against 10 of 16 RBD proteins (significance by ^) including all VoCs except Omicron, as well as bat RaTG13 and pangolin Gx-P5L viruses but not to SARS-CoV-1 and related sarbecoviruses. Coronavac only boosted responses to 8 of 16 RBD proteins (significance by #), but to lower magnitude than BNT162b2. The BNT162b2 post-vaccination sVNT responses were substantially higher than Coronavac in 10 of 16 RBDs (significance by **). Therefore, the overall magnitude of neutralizing antibody responses by Coronavac vaccination was substantially lower than BNT162b2 vaccination and not above the 20% inhibition cut-off for any RBD protein (Fig. 2a).

BNT162b2 vaccination 1 year after recovery from mild ancestral SARS-CoV-2 infection (Fig. 2b) led to very high (mean > 50% inhibition)

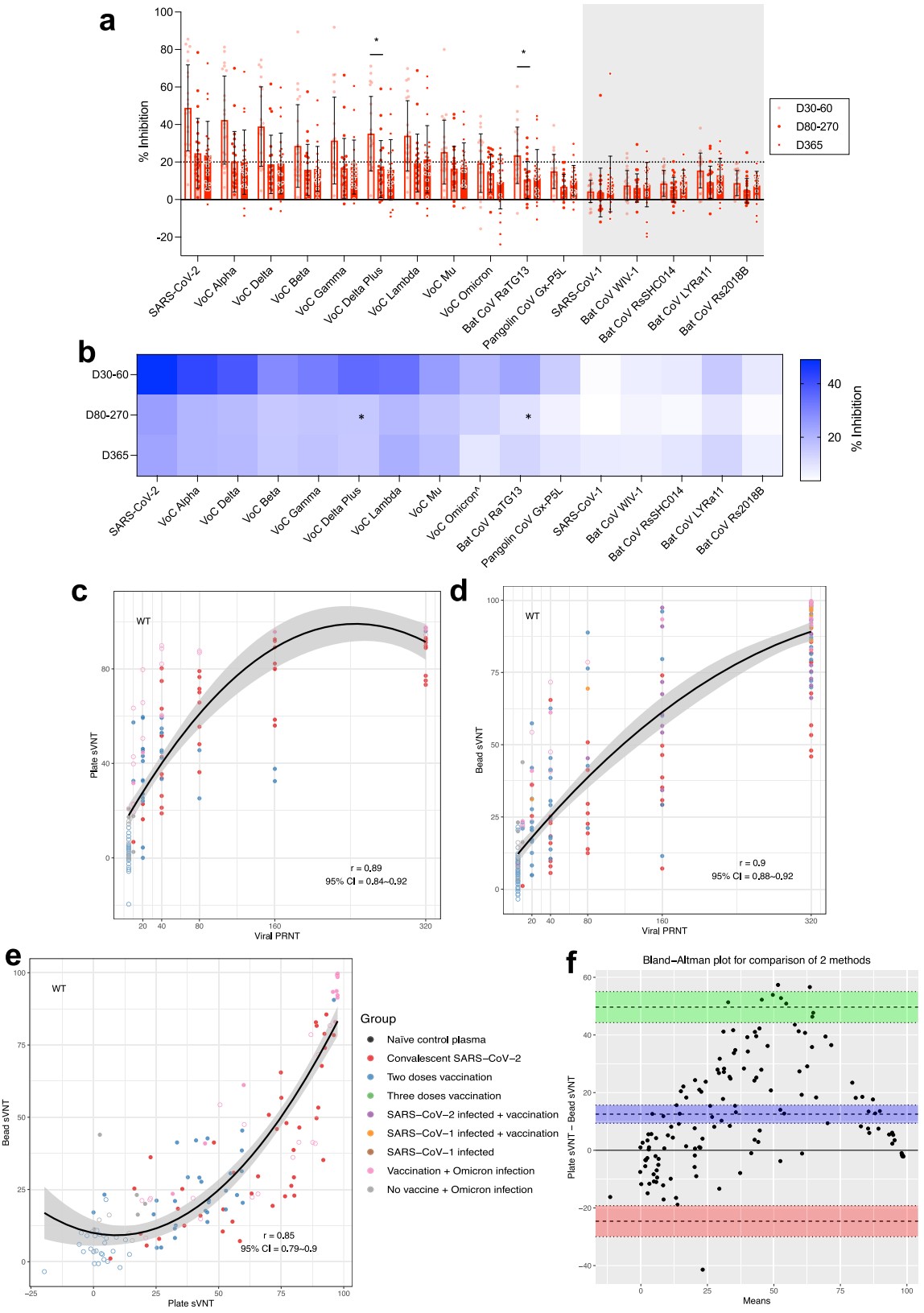

responses against all VoCs including Omicron as well as bat RaTG13 and pangolin Gx-P5L viruses and lower (mean inhibition between 20–50%) against SARS-CoV-1 and other sarbecoviruses. Coronavac also elicited responses above the 20% cut-off against 15 of 16 RBD proteins, but 9 of these (SARS-CoV-2, VoC's Alpha, Delta, Lambda, Bat CoV RaTG13 and Pangolin CoV Gx-P5L) were still significantly lower in

comparison with BNT162b2 vaccination. Thus, prior immune priming with SARS-CoV-2 infection substantially improved the antibody breadth and magnitude of responses to inactivated whole virus vaccines, but not to the same extent as S-specific mRNA vaccination.

Priming by prior exposure to SARS-CoV-1 infection, 18 years prior to BNT162b2 vaccination ($n = 7$) elicited pan-sarbecovirus antibodies

**Fig. 1 | sVNT multiplex assay shows relative ACE2 binding affinity to determine the sarbecovirus antibody profile which correlates with plate and viral based assays. a** The % antibody inhibition of ACE2 binding to RBD was determined for convalescent samples following SARS-CoV-2 infection from day 30–60 ($n = 20$ subjects), day 80–270 ($n = 20$ subjects), day 365 ($n = 22$ subjects), and represented as a heatmap (**b**). ^Omicron sVNT bead-based results, data following uses plate-based results. **a** Dotted line indicates 20% inhibition as a positive result based on limit of quantification, and data represented mean+/− stdev and individual samples shown, grey shading indicates SARS-CoV-1 clade. **a, b** Kruskal-Wallis multiple comparisons test, *$p < 0.05$. Spearman correlation analysis (r) analysis for (**c**) Ancestral wild type SARS-CoV-2 RBD plate-based sVNT versus SARS-CoV-2 infectious virus PRNT50, (**d**) bead based sVNT versus SARS-CoV-2 infectious virus PRNT50, and **e** wild type SARS-CoV-2 RBD plate-based sVNT versus bead based sVNT. **f** Bland Altman analysis of wild type SARS-CoV-2 RBD plate-based sVNT versus bead based sVNT results. **f** The red and green areas show the limits of agreement (Upper limit of agreement: 49.67, Lower limit of agreement: −24.61). **c–e** Spearman correlation analysis (r), data represents the individual data, dotted lines show 95% confidence bands of the best-fit line. Pre-pandemic plasma samples ($n = 30$) were used as negative controls for antibody for inhibition of ACE2 binding of the 16-plex RBD panel for % inhibition as = 100*(Mean FI of 30 negative pre pandemic samples - individual FI)/Mean FI of 30 negative pre pandemic samples. Samples were run in duplicate for the sVNT assay and experiments were repeated twice.

to all RBDs tested, covering both the SARS-CoV-2 and SARS-CoV-1 clades. Coronavac vaccination after SARS-CoV-1 infection ($n = 2$), showed post vaccination responses across the RBD panel (Fig. 2c), but our small samples size precludes statistical comparisons. Vaccination of SARS-CoV-1 recovered individuals in Guangzhou with Sinopharm (Fig. 2d), another inactivated SARS-CoV-2 vaccine used in mainland China, led to responses to SARS-CoV-2 variants (except Omicron), SARS-CoV-1 and related sarbecoviruses but post vaccination rises were only significant in 3 of 16 RBDs (ancestral, VoCs Alpha and Gamma).

The breadth and magnitude of the RBD-specific neutralizing antibody response across these prior infection conditions by heatmap (Fig. 2e), shows limited clade 1 antibodies without prior infection, i.e., in Coronavac and BNTb162b2 vaccination of naïve individuals. Whilst prior COVID-19 with vaccination increases the breadth and magnitude of the RBD antibody response. Furthermore, BNT162b2 vaccination shows higher magnitude responses than inactivated vaccines, especially with historic SARS-CoV-1 infection.

### Third dose mRNA vaccination boosts SARS-CoV-1 clade neutralizing antibody responses

We conducted an observational study of third dose vaccination of Coronavac or BNT126b2, following homologous 2-dose priming with either Coronavac or BNT126b2, resulting in 4 vaccine comparison groups (CC + C, CC + B, BB + C, BB + B) (Fig. 3a, b). The third dose vaccination was given approximately 6 months after the second vaccination. The post third dose BNT162b2 vaccination following either BNT162b2 or Coronavac priming led to substantial boosting of antibody responses across the panel (Fig. 3b) to 14 and 13 of 16 RBDs, respectively. Whilst Coronavac priming followed by a third dose of Coronavac led to significant increases in antibody in 5 of 16 RBDs, the magnitude of these responses was substantially lower than third dose BNT162b2 groups. There was no boosting of neutralizing antibody in those given a third dose of Coronavac following two-dose priming with BNT162b2 (Fig. 3c).

### Antibody responses following Omicron BA.2 infection in vaccinated or naïve individuals

We compared acute (day 0–5 of infection) and convalescent (1–2 months post infection) sera from Omicron infection in those previously naïve or vaccinated with BNT162b2 or Coronavac vaccination (Fig. 4a, b). For BA.2 infection in BNT162b2 vaccinated subjects

### Table 1 | Samples used in sVNT assay

| Group | Time point | Legend | Sample # |
|---|---|---|---|
| SARS-CoV-2 convalescent | 30-60d | D30-60 Rec | 20 |
| SARS-CoV-2 convalescent | 180-270d | D80-270 Rec | 20 |
| SARS-CoV-2 convalescent | 365d | D365 Rec | 22 |
| BNT162b2 | Pre/1 M post | Pre-BB/BB | 30 |
| CoronaVac | Pre/1 M post | Pre-CC/CC | 30 |
| SARS-CoV-2 convalescent + BNT162b2 | 1 M post | SARS2 + B | 20 |
| SARS-CoV-2 convalescent + Coronavac | 1 M post | SARS2 + C | 20 |
| SARS-CoV-1 patient from HK + BNT162b2 | Pre/1 M post | SARS1 + pre B<br>SARS1 + BB | 7<br>7 |
| SARS-CoV-1 patient from HK + Coronavac | Pre/1 M post | SARS1 + pre C<br>SARS1 + CC | 2<br>2 |
| SARS-CoV-1 patient from GZ Sinopharm | 2018, 1 M, 3 M, 6 M post | SARS1 + pre S<br>SARS1 + SS 1 M<br>SARS1 + SS 3 M<br>SARS1 + SS 6 M | 10<br>6<br>5<br>2 |
| Coronavac (2 doses) + Coronavac booster | Pre/1 M post | D0 CC + C/<br>D30 CC + C | 20 |
| CoronaVac (2 doses) + BNT162b2 booster | Pre/1 M post | D0 CC + B/<br>D30 CC + B | 20 |
| BNT162b2 (2 doses) + Coronavac booster | Pre/1 M post | D0 BB + C/<br>D30 BB + C | 20 |
| BNT162b2 (2 doses) + BNT162b2 booster | Pre/1 M post | D0 BB + B/<br>D30 BB + B | 20 |
| Omicron infected (unvaccinated) | Acute/Recovered | No vaxx + Omicron Acute/Rec | 10 |
| Omicron infected vaccinated BNT162b2 | Acute/Recovered | B + Omicron Acute/Rec | 20 |
| Omicron infected vaccinated Coronavac | Acute/Recovered | C + Omicron Acute/Rec | 14 |
| High neut positive control | | | 1 |
| Medium neut positive control | | | 1 |
| WHO standard 20/136 | | | 1 |
| Naïve pre pandemic negative control | | | 30 |

### Table 2 | RBD amino acid sequence homology versus ancestral SARS-CoV-2

| SARS-CoV-2 RBD vs. | % aa conservation | Number aa difference |
|---|---|---|
| VoC Alpha | 99.6 | 1 |
| VoC Delta | 99.1 | 2 |
| VoC Beta | 98.7 | 3 |
| VoC Gamma | 98.7 | 3 |
| VoC Delta Plus | 98.7 | 3 |
| VoC Lambda | 99.1 | 2 |
| VoC Mu | 98.7 | 3 |
| VoC Omicron (BA.1) | 93.3 | 15 |
| Bat CoV RaTG13 | 90.1 | 22 |
| Pangolin CoV Gx-P5L | 86.6 | 30 |
| SARS-CoV-1 | 73.1 | 60 |
| Bat CoV WIV-1 | 75.3 | 55 |
| Bat CoV RsSHC014 | 75.3 | 55 |
| Bat CoV LYRa11 | 73.5 | 59 |
| Bat CoV Rs2018B | 74.9 | 56 |

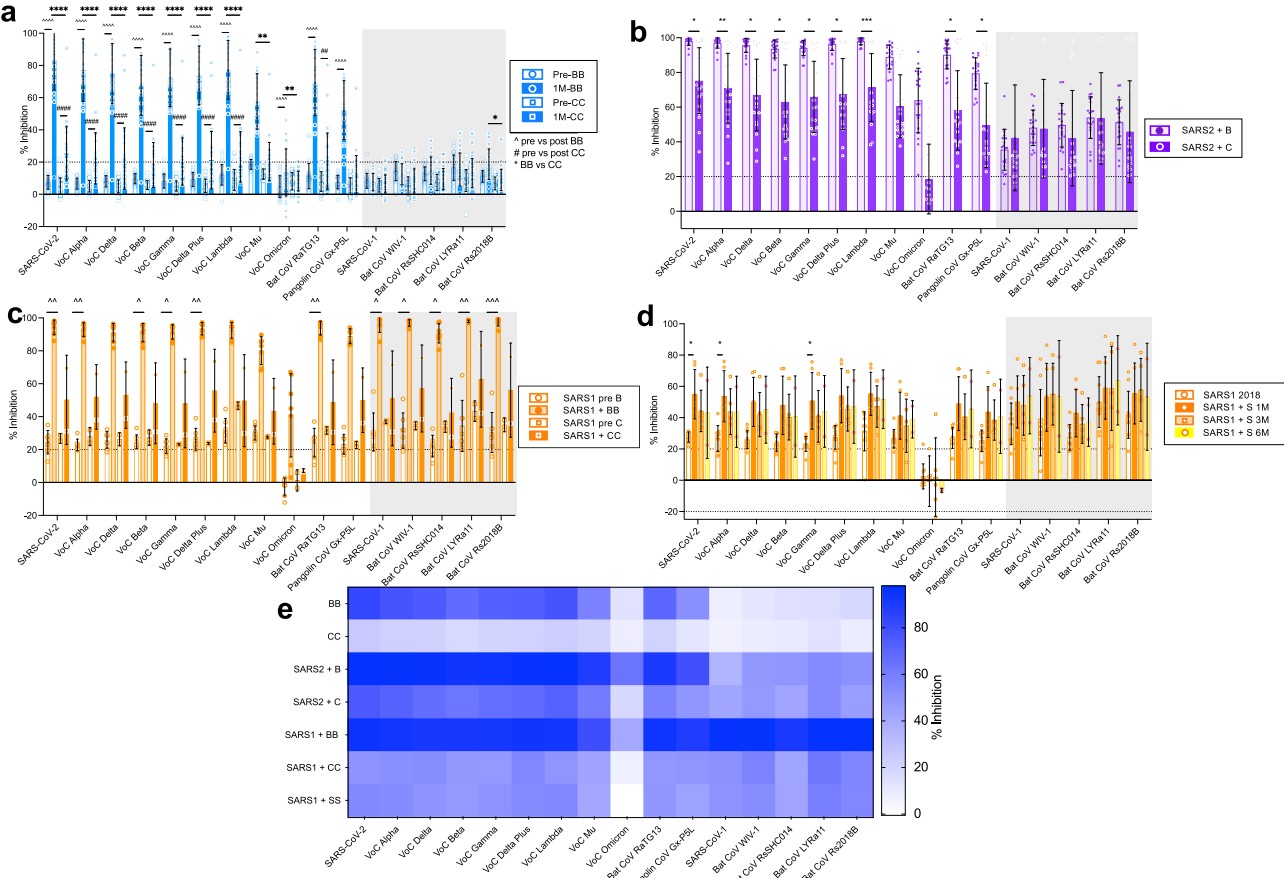

**Fig. 2 | Prior priming by infection accentuates mRNA vaccine antibody profile to sarbecovirus RBD.** The % antibody inhibition of ACE2 binding to RBD was determined for **a** uninfected subjects pre and 1 month post 2 dose vaccination with BNTb162b2 (B) or Coronavac (C) ($n = 30$ subjects). **b** COVID-19 infected 1 month post 1 dose vaccination with BNT162b2 (SARS2 + B) or Coronavac (SARS2 + C) ($n = 20$ subjects, prior infection 346+/−105 days before vaccination). **c** Hong Kong (HK) SARS-CoV-1 infected patients (SARS1) 1 month post 2 dose vaccination with BNT162b2 (SARS1 + B, $n = 7$ subjects) or Coronavac (SARS1 + C, $n = 2$ subjects). **d** Guangzhou (GZ) SARS-CoV-1 infected patients from 2018 ($n = 10$ subjects) and post 2 dose vaccination with Sinopharm (SARS1 + S) at 1 to 6 months post

vaccination (1 M $n = 6$ subjects, 3 M $n = 5$ subjects, 6 M $n = 2$ subjects). **e** Heat map representation of % inhibition of 1 month post vaccination (from **a**–**d**). **a**–**d** Data represented mean+/− stdev and individual samples. **a, c** Significant differences in paired pre versus post vaccine responses within vaccine type by one-way Friedmans tests with Dunns multiple comparisons (coloured, within vaccine type). **b, d** one-way Kruskall Wallis test with Dunns multiple comparisons between vaccine types (**a, b**) or versus 2018 (**d**) (black, statistical differences). *$p < 0.05$, **$p < 0.01$, ***$p < 0.001$, ****$p < 0.0001$, ^ for BNT pre versus post and # for Coronavac pre versus post.

there were significant increases for 6 RBDs of clade 2 SARS-CoV-2 viruses at recovery, with high magnitude (>50% inhibition) across all clade 2 RBDs, and detectable responses (above the 20% cutoff) for clade 1 viruses but not increased from acute timepoints. In addition, Coronavac primed individuals infected with Omicron BA.2, had significant increases in antibody responses to 4 RBDs of clade 2 SARS-CoV-2 viruses at recovery, whilst clade 1 responses were unchanged. In marked contrast, recovery from Omicron BA.2. in those without prior vaccine or infection priming did not lead to increases in antibody responses to any RBDs (excluding VoC Gamma), demonstrating its poor immunogenicity.

## Overview of antigenic diversity from different priming conditions

To provide an overview of our results above, we generated a 2-D representation of all tested samples (Table 1) to determine which priming conditions led to broader and higher magnitude antibody responses to a range of SARS-CoV-2 VoCs (Fig. 5a) and across all sarbecoviruses (Fig. 5b). RBD cross-reactivity is a measure of the antibody response diversity, i.e., the frequencies of responses to different RBD proteins above a 20% inhibition cut-off (see Methods). The priming conditions that yielded higher magnitude (>75% inhibition) and

breadth (>75% cross-reactivity) of SARS-CoV-2 VoC antibody responses (Fig. 5a) included the groups SARS-2 followed by BNT162b2, third doses of BNT162b2 (CC + B and BB + B), SARS-1 with BNT162b2, Omicron BA.2 breakthrough following two doses of BNT162b2 or Coronavac vaccines. Whilst inactivated vaccines, Coronavac and Sinopharm in SARS-1 convalescents led to high antigenic diversity, the magnitude of these responses was not maximized (40–50% inhibition). Thus, third dose vaccination and BNT162b2 vaccination after COVID-19 recovery results in maximal antibody diversity and response magnitude and should be continued to be recommended to increase protection against future VoC.

When both magnitude and breadth of responses to the broader RBD panel including SARS-CoV-1 clade viruses are considered, i.e., true pan-sarbecovirus antibody responses (Fig. 5b), SARS-CoV-1 followed by BNT162b2 vaccination is significantly better than any other condition. Several conditions including SARS-2 followed by BNT162b2 or Coronavac immunization, third dose BNT162b2 following BNT162b2 or Coronavac priming, SARS-CoV-1 followed by Sinopharm or Coronavac immunization, Omicron breakthrough infections in BNT162b2 vaccinated provide high breadth of protection (>75%) across the sarbecovirus group with moderate magnitude of antibody inhibition. Two doses of BNT162b2 or Coronavac, three doses of Coronavac or two

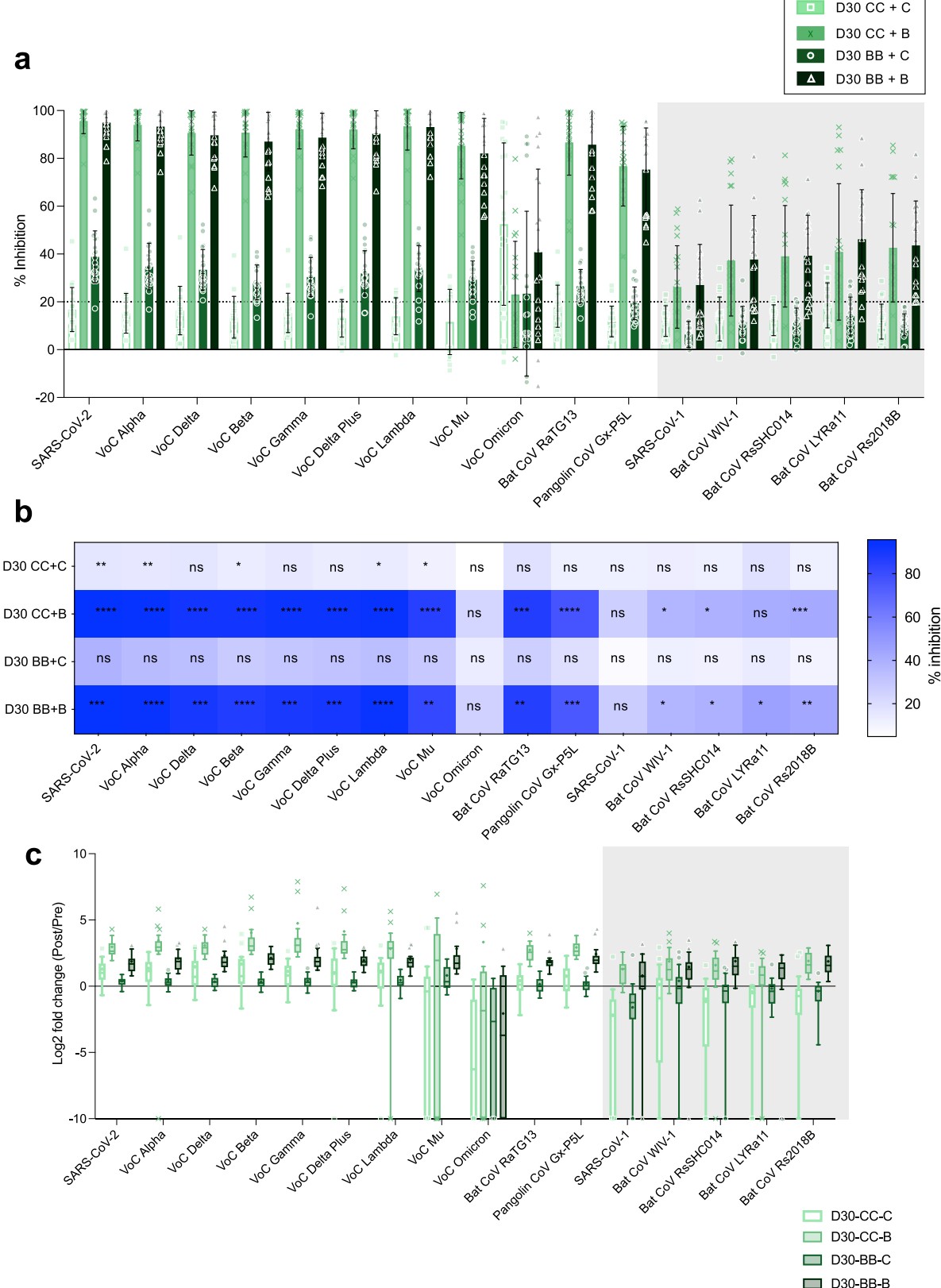

**Fig. 3 | mRNA third dose vaccination regardless of priming increases antibody breadth. a** The % antibody inhibition of ACE2 binding to RBD was determined for third dose vaccination after CC or BB prime for either B or C boost (each group $n = 20$ subjects) 1 month post third dose vaccination. **b** Heat map representation of % inhibition of 1 month post vaccination (from **a**). **c** Fold change of pre versus post booster vaccination (from **a**). **a** Data represented mean+/− stdev and individual samples. **b** Significant differences in paired pre versus post vaccine responses within vaccine type by one-way Friedmans tests with Dunns multiple comparisons. *$p < 0.05$, **$p < 0.01$, ***$p < 0.001$, ****$p < 0.0001$, ns not significant.

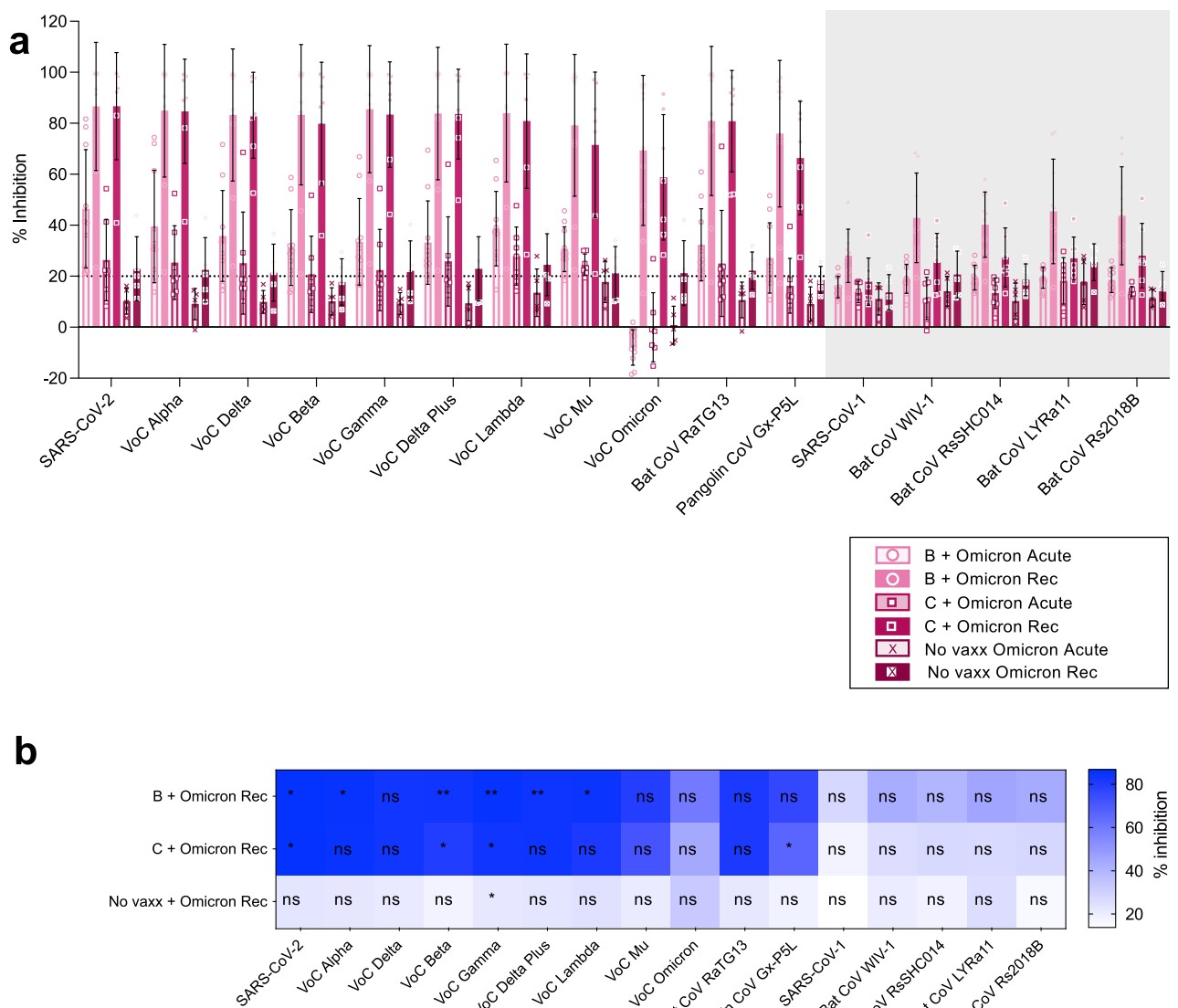

**Fig. 4 | Omicron infection in combination with vaccine priming broadens antibody response.** The % antibody inhibition of ACE2 binding to RBD was determined for **a** BNT162b2 (n=10 subjects), Coronavac (*n* = 7 subjects), unvaccinated (*n* = 5 subjects) paired acute (day 0–5) and recovered (1–2 months after illness) patients for Omicron BA.2 infection. **a** Data represented mean+/− stdev and individual samples. **a, b** Significant differences in paired pre versus post vaccine responses within vaccine type by one-way Friedmans tests with Dunns multiple comparisons. *$p < 0.05$, **$p < 0.01$, ***$p < 0.001$, ****$p < 0.0001$. Statistics shown for acute versus recovered (**b**), and ns between Coronavac and BNT162b2 Omicron recovered samples by Ordinary one-way Anova w Sidak's multiple comparison test.

doses of BNT162b2 followed by Coronavac do not yield antibody with either higher breadth or magnitude.

## Discussion

Optimal strategies for developing and using vaccines that protect against current, and hopefully future, SARS-CoV-2 VoCs ("variant-proof vaccines" COVID-19 vaccines) are a current priority for global public health. Since other sarbecoviruses, not just SARS-CoV-2, continue to pose future pandemic threats, strategies that elicit broad sarbecovirus immune responses also need to be developed. The antibody responses generated by diverse COVID-19 vaccines in naïve individuals or after infection with SARS-1 or VoC Omicron, boosting and heterologous vaccination provides an opportunity to address these challenges. Furthermore, Hong Kong has maintained a rigorous zero-COVID policy with low virus circulation until January 2022 (population based seroprevalence ~1%), and therefore in this study vaccine and infection immunogenicity was in a naïve population or with RT-PCR confirmed infection. A recently described multiplex bead

surrogate neutralization assay provides the opportunity to investigate neutralizing antibody responses to a range of SARS-CoV-2 variants and sarbecoviruses[21,23]. We assembled a panel of plasma samples from individuals with a variety of conditions of priming, boosting and infection histories to investigate how these impact on the breadth and magnitude of neutralizing antibody responses to this broad panel of SARS-CoV-2 VoCs and sarbecoviruses.

We found that the results of the bead-based multiplex sVNT assay correlated well with the plate-based sVNT and with the "gold standard" PRNT assay for the ancestral virus and multiple VoCs, with the exception of Omicron variant BA.1. We therefore used the plate-based sVNT for Omicron VoC for more reliable results that may be due to conformational differences in the stability of Omicron RBD. Many of these sera have been previously tested in PRNT assays using the ancestral virus and Omicron BA.1 and BA.2 VoCs[28,29]. Our findings with the sVNT assays were concordant with previously reported data from PRNT assays; (1) that two dose BNT162b2 vaccination was more immunogenic than two doses of Coronavac but both were poor at

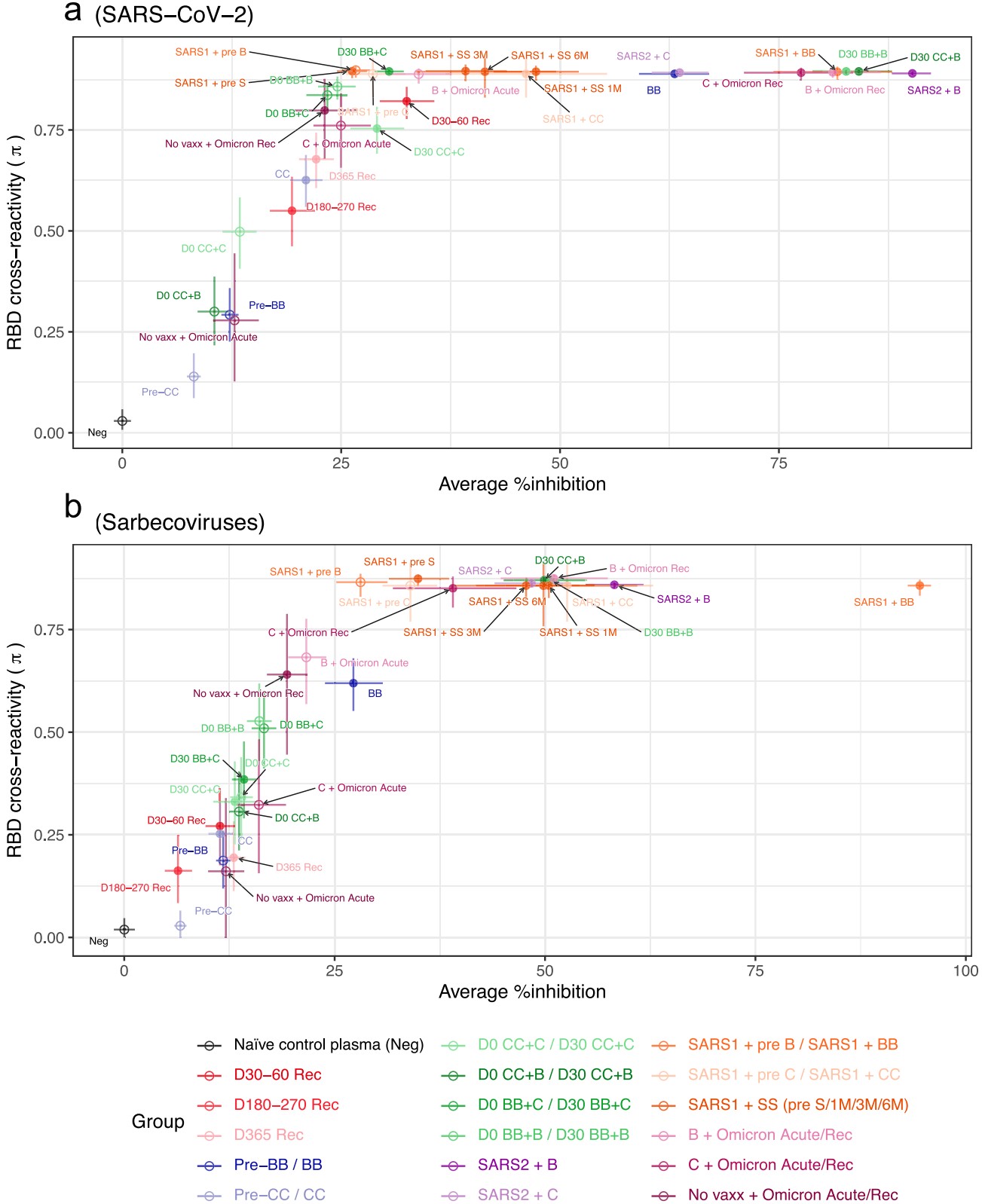

**Fig. 5 | Overview of antigen diversity versus magnitude of neutralizing antibody responses for SARS-CoV2 viruses and sarbecoviruses for different priming conditions.** A 2-D representation of all samples tested (from Table 1) for antigenic diversity versus the average % inhibition of all RBDs for SARS-COV-2 and its VoC (**a**), and all sarbecoviruses tested (**b**) for different priming conditions (from Figs. 2–4). Data represents the group average (95% CI), non-overlapping confidence intervals indicate significantly different responses. Open circles represent baseline samples at pre vaccination, pre third dose vaccination or acute infection samples. Closed circles represent post vaccination or post infection samples at recovered (Rec) timepoints.

eliciting neutralizing antibody to VoC Omicron BA.1[28,29]; (2) a third dose of BNT162b2 following two doses of either BNT162b2 or Coronavac elicited neutralizing antibody to Omicron BA.1 while three doses of Coronavac failed to do so[28]; (3) that hybrid immunity elicited by a single dose of BNT162b2 in those previously convalescent from ancestral SARS-CoV-2 infection elicited broader neutralizing antibody responses across the SARS-CoV-2 VoCs including Omicron, to levels at least comparable to three doses of BNT162b2[28,29]; and (4) that Omicron BA.2 breakthrough infections in BNT162b2 or Coronavac vaccinated individuals elicited broad neutralization of all VoCs tested in either PRNT or sVNT assays[29]. The multiplex sVNT assay further demonstrated that cross reactivity generated by these vaccines and hybrid immunity against the SARS-CoV-2 VoC, extended to the related bat RaTG13 and pangolin Gx-P5L viruses, even better than to Omicron VoC, but not necessarily to the more distantly related SARS-CoV-1 and sarbecoviruses.

We devised a 2D depiction where both the breadth and magnitude of sVNT neutralizing antibody responses could be visualized. The greatest breadth and magnitude neutralization activity across the SARS-CoV-2 and SARS-CoV-1 related sarbecoviruses was elicited by those recovered from SARS-CoV-1 infection in 2003 vaccinated with two doses of BNT162b2, the weakest, though moderate neutralizing activity among all 16 virus RBDs being to Omicron BA.1. This suggests that prime and boost with antigenically diverse sarbecoviruses provided the optimal breadth and magnitude of neutralizing activity. Since Omicron appears to be the most antigenically divergent RBD[23], one may speculate that individuals primed with an ancestral SARS-CoV-2 antigen (e.g., BNT162b2) boosted with an effective Omicron-spike vaccine or a bivalent Omicron and SARS-CoV-1 vaccine may lead to comparable or superior breadth of immunity. However, the number of RBD mutations does not define neutralizing antibody escape, and Omicron represents a challenge in being more genetically similar but antigenically distant to ancestral SARS-CoV-2, than other clade 2 viruses Bat CoV RaTG13 and pangolin CoV GX-P5L[23]. Furthermore, recovery from Omicron infection in Coronavac and BNT162b2 vaccinated individuals, generated antibody breadth but did not maximize response magnitude to the same extent as SARS-CoV-1 infection. Similarly, recent SARS-CoV-2 infection then subsequent BNT162b2 vaccination generated greater SARS-CoV-2 clade antibody responses than prior SARS-CoV-1 infection, however these responses were not maximized in terms of magnitude either, which could also be attributable to one dose versus two dose vaccination respectively. Therefore, either SARS-CoV-1 represents an antigenic 'sweet spot' for generating broad antibody responses, or recall of a long term memory B cell response, from infection 18 years prior, adds to response magnitude capacity. This could be exploited by mosaic vaccines or heterologous prime boost approaches.

High level of cross-reactivity was elicited by hybrid immunity involving ancestral or Omicron SARS-CoV-2 infections and BNT162b2 vaccination, although the magnitude of neutralizing activity was less than that following SARS-CoV-1 infection. A third dose of BNT162b2 also elicited notable breadth of cross-neutralizing activity across the sarbecovirus group, even in Coronavac primed subjects. It remains to be seen whether the durability of this magnitude and breadth following three doses of BNT162b2 vaccination alone is similar to that elicited by hybrid immunity. Infection priming with different VoC (prior to Omicron) and subsequent Ancestral SARS-CoV-2 vaccination would likely generate similar breadth of pan-sarbecovirus responses due to residual cross-reactivity[30,31] and the overwhelming titer of nAb that are generated. The vaccine type may outweigh the VoC prime, however this remains to be determined experimentally for various VoC as imprinting does occur, it does not occlude new antibody responses. Furthermore, it remains to be determined if the high cross-reactivity of BNT162b2 responses generated 1 month after vaccination are an artefact of high magnitude of the plasm blast antibody response, and

what response is actually functionally recalled in vivo during infection months or years later from memory B cells. This applies to both durability of circulating antibody as well as memory B cell responses because rapid recall of memory B cell responses may well compensate for a fall in circulating antibody levels. It is notable that in those convalescent from ancestral SARS-CoV-2 infection, there was a gradual decline of both breadth and magnitude of neutralizing responses over time.

Our study had some limitations in testing for antibodies outside of the RBD or other immune correlates, and other vaccine platforms. First, our neutralization assay only assesses neutralizing activity directed to the RBD and does not assess neutralizing activity directed to other known regions of the S protein, including the N terminal domain (NTD), the S2 domain, or S-ecto domains[32,33]. It is however worth noting that the majority of SARS-CoV-2 neutralizing antibodies target the RBD region, as deletion of NTD antibodies have minimal impact on neutralization[34]. sVNT assays are variant dependent, and Omicron RBD antibodies were assessed by both bead and plate approaches. Results are shown for the plate-based method due to better correlation with gold standard PRNT results. All other RBDs were represented in the multiplex assay simultaneously. Secondly, we have focused on neutralizing activity, which is the only known correlate of protection so far[1,35]. But it is likely that T cell responses[8] and non-neutralizing antibody and their effector functions[9,10] also contribute to protection against severe disease, but are not covered in our study. Thirdly, we have only compared RNA vaccines and inactivated vaccines but not assessed other vaccine strategies such as the adenovirus vectored vaccines. However, these 2 vaccines represent distinct platforms known to induce neutralizing antibodies at the high and low ends of the antibody response range, respectively, and are most widely used vaccines globally. Recently, a related study has investigated booster vaccines, including viral vectored AZD1222[23], with a similar reporting of mRNA vaccine advantage for antibody breadth.

## Methods

### Study participants for serum panels

Plasma panels from cohorts with different combinations of vaccination and natural infection were used in the study (Table 1, see Supplementary data 1 for subject demographics, age 49+/−13 years, range 21–77 years). Pre-pandemic plasma samples ($n = 30$ subjects) were used as negative controls for antibody for inhibition of ACE2 binding of the 16-plex RBD panel and used to calculate % inhibition for each RBD. To assess antibody waning and breadth, RT-PCR confirmed SARS-CoV-2 convalescent samples obtained from individuals with infection occurring in the period January to March 2020 when the ancestral SARS-CoV-2 was circulating were used, with samples collected at day 30–60 ($n = 20$ subjects), day 80–270 ($n = 20$ subjects), day 365 ($n = 22$ subjects) post infection, with no further vaccination or infection during sampling. This corresponds to the SARS-CoV-2 used in the vaccine, RBD panel and PRNT assay.

To assess vaccine immunogenicity, plasma was collected from previously uninfected subjects (confirmed by N ELISA), prior to receiving the first vaccine dose and at 1 month post 2-dose vaccination with BNT162b2 or Coronavac ($n = 30$ subjects). BNT162b2 vaccination is recommended 21 days apart and Coronavac 28 days apart. To assess the impact of prior infection (hybrid immunity), subjects ($n = 20$ subjects) who recovered from SARS-CoV-2 infection (346+/−105 days between SARS-CoV-2 infection and vaccination), were assessed 1 month after 1 dose vaccination with BNT162b2 or Coronavac. Participants with prior exposure to SARS-CoV-1 in 2003 were recruited in Hong Kong (HK) and sampled pre- and 1 month post 2-dose vaccination with BNT162b2 ($n = 7$ subjects) or Coronavac ($n = 2$ subjects). In addition, serum from SARS-CoV-1 infected patients in Guangzhou (GZ), with 'baseline' serum from 2018 ($n = 10$ subjects) and post 2-dose vaccination with Sinopharm were sampled at 1 month

($n$ = 6 subjects), 3 month ($n$ = 5 subjects) and 6 months ($n$ = 2 subjects) post vaccination.

To assess the impact of heterologous third dose vaccination for either BNT162b2 or Coronavac, individuals were randomized 3 months after 2-doses of vaccination with either Coronavac (CC) or BNT162b2 (BB) to receive a third dose of Coronavac (BB + C or CC + C) or BNT162b2 (CC + B or BB + B), and samples collected at pre-third dose and 1 month post third dose vaccination (group and timepoint, $n$ = 20 each).

Omicron infection (BA.2 predominant strain at time of serum collection in Hong Kong January-February 2022) of participants (50+/ −17 years of age) with prior vaccination of BNT162b2 ($n$ = 10 subjects), Coronavac ($n$ = 7 subjects) (some donors are 1, 2 or 3 dose vaccinated), unvaccinated ($n$ = 5 subjects) with paired acute (day 0−5) and recovered (1−2 months after illness) samples were tested. Plasma was separated from venous blood and stored at −80 °C and heat inactivated at 56 °C for 30 min prior to use.

The study was approved by the Joint Chinese University of Hong Kong-New Territories East Cluster Clinical Research Ethics Committee (Ref no.: 2020.229), the First Affiliated Hospital of Guangzhou Medical University Ethics Committee (Ref no: 2018.044). The third dose vaccine study protocol was approved by the Institutional Review Board of the University of Hong Kong (Ref: UW 21-492), and the Clinicaltrials.gov registration number is NCT05057169, subjects were randomised for their third dose vaccination after homologous 2-dose vaccination with either BNT162b2 or Coronovac[36]. For each study cohort all participants provided written informed consent.

### Sarbecovirus RBDs for the 16-plex sVNT assay system

A 16-plex RBD panel of biotinylated proteins was prepared[23]. Briefly, The RBDs included in this study are as follows: (A) Clade-2 sarbecoviruses: SARS-CoV-2 Ancestral (Wuhan-hu-1), SARSCoV-2 VoCs (Alpha, Beta, Gamma, Delta, Omicron), SARS-CoV-2 variants of interest (Delta plus, Lambda, Mu), Bat CoV RaTG13, Pangolin CoV GX-P5L; (B) Clade-1 sarbecoviruses: SARS-CoV-1 and bat CoVs WIV-1, Rs2018B, LYRa11 and RsSHC014. Proteins were custom made (SARS-CoV-2, SARS-CoV-2 Alpha, Delta, Beta, Gamma, Bat CoV RaTG13, Pangolin CoV GX-P5L and SARS-CoV-1 were custom made by Genscript), purchased (Omicron RBD, Acrobiosystems), or produced in house (SARS-CoV-2 Delta plus, Mu and Lambda, Bat CoVs WIV1, Rs2018B, LYRa11 and RsSHC014) in HEK293T cells, from ATCC. RBD proteins were enzymatically biotinylated and coated on MagPlex-Avidin microspheres (Luminex) at 5 μg RBD protein per 1 million beads for use in the sVNT assay.

RBD-coated beads (25 μl, 600 per antigen) were pre-incubated with 25 μl heat inactivated serum at 1:20, for 15 min at 37 °C with agitation (200 rpm), followed by addition of 50 μl of PE conjugated human ACE2 (2 mg/ml; Genscript) and incubated for an additional 15 min at 37 °C with agitation. After two washes with 1% BSA in 1 M NaCl PBS, the final readings were acquired using the MAGPIX system (Luminex, array reader v2.6.1, microplate platform v2.1.15, Bio-Plex manager software v6.2.0.175) following manufacturer's instruction.

To assess surrogate virus neutralisation the MFI of each RBD bead region was used to calculate: % inhibition = 100*(Mean FI of 30 negative pre pandemic samples−individual FI)/Mean FI of 30 negative pre pandemic samples. Percentage inhibition > =20% is typically considered as positive for SARS-CoV-2 neutralizing antibody, while percentage inhibition <20 was considered as negative[21], as indicated by dotted lines at 20% on most figures, however sVNT results are shown for all samples including those that are lower than pre-pandemic controls, resulting in some negative values.

### Plate based sVNT commercial assay

For the ancestral and Omicron BA.1 RBD proteins for SARS-CoV-2 surrogate virus neutralization test[27] (sVNT) kits were used (Cat. No.: L00847-A and Z03728, GenScript, Inc., NJ, USA). The tests were

performed according to the manufacturer's standard protocol. Samples, positive and negative controls were 10 times diluted and then mixed with equal volume of horseradish peroxidase (HRP) conjugated SARS-CoV-2 spike receptor binding domain (RBD) (6 ng). The mixture was then incubated at 37 °C for 30 min. After incubation, 100 ul of the mixture was added to corresponding wells of the capture plate coated with ACE2 receptor. The plate was sealed and incubated at 37 °C for 30 min. The plated was then emptied and washed with 1X wash solution for 4 times. Residual liquid was removed by tapping dry. 100 μl of TMB solution was added to each well and the plate was wrapped with aluminium foil and incubated in the dark at room temperature for 15 min. The reaction was quenched by adding 50 μl of stop solution. The absorbance was read at 450 nm ($OD_{450}$) in an ELISA microplate reader. To assess surrogate virus neutralisation the $OD_{450}$ was used to calculate: % inhibition = 100*(1- $OD_{450}$ value of sample/$OD_{450}$ value of negative control)

### PRNT assay

The PRNT was performed in duplicate using culture plates (Techno Plastic Products AG, Trasadingen, Switzerland) in a Biosafety level 3 facility. Serial serum dilutions from 1:10 to at least 1:320 were incubated with ~30 plaque-forming units of SARS-CoV-2 BetaCoV/Hong Kong/VM20001061/2020 virus for 1 h at 37 °C. The virus-serum mixtures were added on to Vero-E6 cell monolayers (from ATCC) and incubated for 1 h at 37 °C in a 5% $CO_2$ incubator. The plates were overlaid with 1% agarose in cell culture medium and incubated for 3 days when the plates were fixed and stained. Antibody titres were defined as the reciprocal of the highest serum dilution that resulted in >90% (PRNT90, a more stringent cut-off) or >50% (PRNT50) reduction in the number of plaques. Values below the lowest dilution tested (1:10) were imputed as 5 and those above 320 were imputed as 640.

### Statistical analysis

Statistical analysis was performed using GraphPad Prism v9 software. Statistically significant differences in paired pre- versus post- vaccine responses within vaccine type were determined by one-way Friedmans tests with Dunns multiple comparisons (coloured*). For comparisons between vaccine groups, a one-way Kruskall Wallis test with Dunns multiple comparisons (black *) was used. Symbols denote statisitical comparisons. ^ for BNT162b2 and # for Coronavac, and * for between vaccine or timepoint comparisons. *, # or ^ = $p < 0.05$, **, ## or ^^ = $p < 0.01$, ***, ### or ^^^ = $p < 0.001$, ****, #### or ^^^^ = $p < 0.0001$, ns = not significant. Correlations between sVNT and PRNT were analysed using Regression analysis toolpack (Excel v16.64).

For RBD cross-reactivity we measured the breadth of antibody responses against different RBDs by calculating the RBD cross-reactivity ($\pi$) in this study. The concept of RBD cross-reactivity ($\pi$) is borrowed from nucleotide diversity ($\pi$) which provides an unbiased estimate of diversity among groups[37]. Specifically, the frequencies of positive RBD responses (the number of RBD responses above a 20% inhibition cut-off) were summarized for each RBD/group, and all the negative responses were characterized in a negative group. Then for every group, where $n_i$ samples of RBD/negative responses $i$ are observed, RBD cross-reactivity ($\pi$) can be calculated based on pairwise difference between antigens (RBD/negative groups) as

$$\pi = \frac{\sum_{i \neq j} n_i n_j}{\frac{1}{2} N(N-1)}$$

where $N$ is the total number of all responses. We also calculated the classic Shannon entropy for comparison and the results are comparable, detailed implementation of the diversity measurement can be found via https://github.com/Leo-Poon-Lab/SARS-CoV-2-sVNT-diversity[38].

For correlation between bead sVNT, plate sVNT and viral PRNT responses, we calculated the spearman correlation between bead sVNT, plate sVNT and viral PRNT responses when paired data were available. The regression line shown in figure was approximated by local polynomial regression fitting with span of 10, the corresponding 95% confidence intervals were shown in grey area.

For confidence intervals for estimating the uncertainty of estimates of RBD cross-reactivity, average %inhibition of responses, and spearman correlation coefficients, the 95% confidence intervals (bootstrap percentile intervals) were estimated using bootstrap resampling of 10,000 times.

### Reporting summary

Further information on research design is available in the Nature Research Reporting Summary linked to this article.

## Data availability

The raw data that support the findings of this study included as supplementary data 1 (Subject details) and source data (experimental data). Source data are provided with this paper.

## Code availability

Source codes for antigenic diversity measures are accessible via https://github.com/Leo-Poon-Lab/SARS-CoV-2-sVNT-diversity[38].

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

## Acknowledgements

This research was supported by grants from the Health and Medical Research Fund Commissioned Research on the Novel Coronavirus Disease (COVID-19), Hong Kong SAR (COVID1903003: C.K.P.M., M.P. and D.S.H., COVID190115: SAV, COVID19F09: B.J.C. and N.H.L., COVID190126: M.P.), Guangdong-Hong Kong-Macau Joint Laboratory of Respiratory Infectious Disease (20191205) (C.K.P.M.), US National Institutes of Health (Contract no. HHSN272201400006C) (M.P.), the Theme-based Research Scheme of the Research Grants Council of the Hong Kong Special Administrative Region, China (LLMP: T11-705/21-N, B.J.C.: T11-712/19-N), Singapore National Medical Research Council (MOH-000535/MOH-OFYIRG19nov-0002, C.W.T., and COVID19RF-003 and COVID19RF-060, L.W.), National Natural Science Foundation of China (82025001, J.Z.).

## Author contributions

Conceptualization: S.A.V., L.W., M.P. Experimental: J.Z.J., C.W.T., S.M.S.C., A.Y.Y.Y. Specialized data analysis: H.G., L.L.M.P. Sample acquisition: S.M.S.C., C.K.P.M., D.S.H., Y.W., J.Z., N.H.L., B.J.C. Funding acquisition: S.A.V., M.P., L.L.M.P., N.H.L., D.S.H. Project administration: J.Z.J., S.M.S.C. S.A.V., M.P. Supervision: S.A.V., M.P. Writing – original draft: S.A.V., M.P. Writing – review & editing: J.Z.J., S.A.V., M.P., L.L.M.P., L.F.W., C.W.T.

## Competing interests

CWT and L-FW are co-inventors of the surrogate virus neutralization test commercialized by GenScript under the trade name cPass. Other authors declare no conflicts of interest.

## Additional information

[1]HKU-Pasteur Research Pole, School of Public Health, Li Ka Shing Faculty of Medicine, The University of Hong Kong, Hong Kong SAR, China. [2]Duke-NUS Medical School, National University of Singapore, Singapore, Singapore. [3]School of Public Health, The University of Hong Kong, Hong Kong SAR, China. [4]The Jockey Club School of Public Health and Primary Care, The Chinese University of Hong Kong, Hong Kong SAR, China. [5]Li Ka Shing Institute of Health Sciences, Faculty of Medicine, The Chinese University of Hong Kong, Hong Kong SAR, China. [6]State Key Laboratory of Respiratory Disease, National Clinical Research Center for Respiratory Disease, Guangzhou Institute of Respiratory Health, First Affiliated Hospital of Guangzhou Medical University, Guangzhou, China. [7]WHO Collaborating Centre for Infectious Disease Epidemiology and Control, School of Public Health, Li Ka Shing Faculty of Medicine, The University of Hong Kong, Hong Kong SAR, China. [8]Centre for Immunology and Infection (C2i), Hong Kong Science Park, Hong Kong SAR, China. [9]Department of Medicine and Therapeutics, The Chinese University of Hong Kong, Hong Kong SAR, China. [10]Stanley Ho Centre for Emerging Infectious Diseases, Faculty of Medicine, The Chinese University of Hong Kong, Hong Kong SAR, China. [11]Department of Microbiology and Immunology, Peter Doherty Institute for Infection and Immunity, University of Melbourne, Melbourne, VIC, Australia. [12]These authors Contributed equally: Janice Zhirong Jia, Chee Wah Tan. [13]These authors jointly supervised this work Linfa Wang, Malik Peiris, Sophie A Valkenburg. ✉e-mail: sophie.v@unimelb.edu.au

