## [Peer Review File · Nature Communications]

Priming conditions shape breadth of neutralizing antibody responses to sarbecovirusesREVIEWER COMMENTS

Reviewer #1 (Remarks to the Author):

This article provides new and greatly needed data regarding the impacts of homologous and heterologous COVID vaccine immunizations and how these interface with natural infections. This type of study is highly significant in our current state of diversified vaccine platforms and repeated cycles of infection due to Variants of Concern (VOC). Notably, the authors have assembled a unique and highly diverse panel of vaccinated and/or infected patient samples. This unique collection allows them to probe a wide variety of vaccine platform combinations and vaccine-infection combinations. The authors study these samples for viral neutralizing potency against past, current, and potentially future sarbecoviruses. They can conclusively define scenarios that impact breadth and neutralizing potency. There also are combinations that lead to greater potential durability. In all, these data are consistent with other studies in the literature. However, this work expands upon our current knowledge and adds a level of complexity. It is excellently written and very clear. The methodology used is well described and justified. This will lend itself to future reproducibility. While I have a few minor issues listed below, I support the publication of this timely and well-executed study.

Minor comments:

- Line 84. Change "It is ..." to "It contains ...".
- Lines 85-89. Very complex sentence. Please rewrite it into multiple, more straightforward statements.
- Lines 89-93. Very complex sentence. Please rewrite it into multiple, more straightforward statements.
- Line 114. "It is evident we cannot "boost our way" out ...".
 - o I am not sure exactly what this phrase means. It should be written in more defined scientific terms. In fact, this article suggests there are boost strategies to increase the breadth and efficacy of vaccination.
- Results section. Early in the Results section, there needs to be a more explicit definition of the principal assays (multiplex and plate sVNT) as surrogate neutralization assays. I believe this is stated in the Methods, but not Results. Firstly, the abbreviation sVNT needs to be defined at its first appearance within the Results. Secondly, there should be a more explicit statement of the authors' intent to bridge authentic virus assays to the surrogate assays. Both of these goals could be accomplished in lines 153-156 as follows:
 - o "Live virus neutralisation strongly correlated with surrogate virus neutralisation assays (sVNT) in both plate ($r=0.89$) and bead formats ($r=0.9$) (Figure 1cd). These plate and bead sVNT assays were also well correlated for the ancestral virus ($r=0.85$, Figure 1e), VoC Beta ($r=0.83$) and Delta ($r=0.76$) (Supplementary figure 1ab).

Reviewer #2 (Remarks to the Author):

Jia et al analysed cross-reactive neutralising responses against SARS-CoV-2 variants of concerns (VOC), SARS-CoV-2 related bat and pangolin viruses, SARS-CoV-1 and related bat sarbecoviruses. The authors analysed the magnitude and breadth of neutralising responses in sera from individuals vaccinated with two or three doses of BNT162b2 mRNA vaccine or inactivated SARS-CoV-2 vaccines (Coronavac or Sinopharm) with or without a history of previous SARS-CoV-2 or SARS-CoV-1 infection. They used a multiplex surrogate neutralisation assay based on receptor-binding domain. They found that hybrid immunity led to a high level of cross-reactive neutralising responses. High level of cross-reactive neutralising responses to VOC were also observed in individuals

pre-infected with SARS-CoV-2 then vaccinated with BNT162b2. However, a previous infection with SARS-CoV-1 followed by BNT162b2 vaccination led to the highest magnitude of neutralising activity. Cross-reactive neutralising responses were also measured in individuals who experienced Omicron BA.2 breakthrough infection following vaccination with BNT162b2 vaccine and in individuals who received a third dose of BNT162b2 vaccine after 2 doses of the same vaccine or an inactivated vaccine.

This paper is relevant and well written. It is essential to determine how broad neutralising response can be elicited in order to lead a broad protection against future SARS-CoV-2 VOC and/or potential future pathogenic sarbecoviruses. The multiplex surrogate neutralisation assay looks appropriate and the limitations are well described. The strong point in this study is to have a few sera from individuals previously infected with SARS-CoV-1. The weakness of the paper is the figures. They are very difficult to understand because they are very busy, especially these ones showing the percentage of inhibition. The authors should show clear main figures to share their take home messages. Details can be added to supplementary figures.

Major comments:

1) It would great to include a supplementary figure showing the differences in RBD amino acids between SARS-CoV-2, SARS-CoV-1, bat and pangolin-related viruses studied in this paper. It would be a good reminder for the reader. It might help for the interpretation of the results.

2) The breadth of neutralising responses might also be impacted by the type of SARS-CoV-2 strain which infected people before vaccination. In this paper the authors focused on individuals pre-infected with the ancestral strain. Do the authors have access to samples from individuals pre-infected with VOC ? If so, it would be valuable to include these data in comparison.

3) How did the authors exclude asymptomatic infections to SARS-CoV-2 which could impact on neutralising responses ? Did they measure anti-N ? A significant change in anti-N response between 2 timepoints could suggest asymptomatic infection. If so, it would be relevant to include these data in Supplementary data.

4) All figures showing the percentage of inhibition are too busy. I am wondering if the authors should only show the heat maps as main figures. Or they should split their panels showing the percentage of inhibition. In addition, sometimes 2 timepoints are shown in the same column of their histograms it is very confusing for the reader. Could the authors show one timepoint per column ? The slight difference in shape or color between each condition in the percentage of inhibition panels is also confusing. The authors should use a clear and comprehensive panel of colors and shapes.

5) Could the authors mention the dosing interval between the 2 first doses of vaccine (mRNA or inactivated) ? The dosing interval may impact on neutralising responses so it is relevant to have this piece of information in methods.

Minor comment:

VoC or VOC: the authors should make sure they are consistent with abbreviations.

Reviewer #3 (Remarks to the Author):

The manuscript by Jia, et al. describes the use a neutralization-surrogate assay to access the breadth of antibody responses following different sequences of exposure to coronavirus antigen. The study shows, in part, that BNT162b2 vaccination following

prior exposure to SARS-CoV-1 led to antibody responses that were highly neutralizing against other sarbecoviruses including viruses of pandemic potential. Interestingly, this sequence appears to have resulted in better antibody responses against zoonotic strains than the Omicron BA.1 variant. The authors also show that in their hands, boosting with BNT162b2 improved antibody responses compared to boosting with CoronaVac. This was particularly true for individuals who received a two dose series of BNT162b2 as they did not notably increase their antibody response following vaccination with CoronaVac. The broadest and strongest antibody responses were seen in individuals vaccinated with either BNT162b2 or CoronaVac and then boosted with BNT162b2. While there are several novel findings in this manuscript and researchers and clinicians would find the information presented in this manuscript useful, several concerns should be addressed before publication. The methods section, and linked manuscripts, provide a clear path for anyone looking to repeat this work.

1. While well written, overall, the manuscript would benefit from additional editing. For example, line 84 beings, "It is about...", and I assume it should read, "It has about...". Additionally, line 110 should likely read, "...which led to an 11x increase...".
2. In general, and especially in figure two, the text on the figures and the size of the points on the graphs are sometimes too small to reliably resolve. This may just be an artifact of the format this reviewer used to view the figures.
3. The correlation between the assays in figure 1 appears robust, the authors should perform (if not already done) and report the results of a Bland-Altman analysis to ensure that any systemic biases between the assays are accounted for.
4. While understandable given the limitations of the various assays, it remains concerning that a different assay platform was used for the Omicron BA.1 results. This should be discussed more clearly in the limitations section.
5. In the limitations section, the authors should also point out that their primary assays only look at spike and therefore ignore any potential neutralization/protective antibodies against other virus proteins.

REVIEWER COMMENTS

Reviewer #1 (Remarks to the Author):

This article provides new and greatly needed data regarding the impacts of homologous and heterologous COVID vaccine immunizations and how these interface with natural infections. This type of study is highly significant in our current state of diversified vaccine platforms and repeated cycles of infection due to Variants of Concern (VOC). Notably, the authors have assembled a unique and highly diverse panel of vaccinated and/or infected patient samples. This unique collection allows them to probe a wide variety of vaccine platform combinations and vaccine-infection combinations. The authors study these samples for viral neutralizing potency against past, current, and potentially future sarbecoviruses. They can conclusively define scenarios that impact breadth and neutralizing potency. There also are combinations that lead to greater potential durability. In all, these data are consistent with other studies in the literature. However, this work expands upon our current knowledge and adds a level of complexity. It is excellently written and very clear. The methodology used is well described and justified. This will lend itself to future reproducibility. While I have a few minor issues listed below, I support the publication of this timely and well-executed study.

Minor comments:

- **Line 84. Change "It is ..." to "It contains ...".**

From: "It was first reported in November 2021. It is about 50 non-silent mutations and over 2/3 of these mutations are in the spike domain ²."

Updated to: "It was first reported in November 2021, **and contains** about 50 non-silent mutations and over 2/3 of these mutations are in the spike domain "

- **Lines 85-89. Very complex sentence. Please rewrite it into multiple, more straightforward statements.**

From: "In 2002, SARS-CoV (herein called SARS-CoV-1) emerged and caused 8,000 infections with public health measures abating the outbreak ³, unlike SARS-CoV-2, which since 2019 has infected over 500 million people within 24 months, countered by over 12 billion doses of highly effective COVID-19 vaccines have been given to mitigate the impact of SARS-CoV-2."

Updated to: "**In 2002, SARS-CoV (herein called SARS-CoV-1) emerged and caused 8,000 infections with public health measures abating the outbreak ³. In contrast, SARS-CoV-2, has infected over 500 million people within 24 months, and the severity of the pandemic countered by over 12 billion doses of highly effective COVID-19 vaccines.**"

- **Lines 89-93. Very complex sentence. Please rewrite it into multiple, more straightforward statements.**

From: "The most predominantly used SARS-CoV-2 vaccines, are inactivated whole virion adjuvanted vaccines (e.g. Coronavac), which have been more widely administered due to ease, scalability and lower cost of production and relative thermostability in contrast to the second most used vaccine, Spike encoding mRNA vaccines ⁴."

Updated to: "**The most predominantly used SARS-CoV-2 vaccines, are inactivated whole virion adjuvanted vaccines (e.g. Coronavac), which have been more widely administered due to ease, scalability and lower cost of production and relative thermostability. The second most widely used vaccines are Spike encoding mRNA lipoparticle vaccines ⁴⁴.**"

• Line 114. "It is evident we cannot "boost our way" out ...".

o I am not sure exactly what this phrase means. It should be written in more defined scientific terms. In fact, this article suggests there are boost strategies to increase the breadth and efficacy of vaccination.

Sentence deleted.

• Results section. Early in the Results section, there needs to be a more explicit definition of the principal assays (multiplex and plate sVNT) as surrogate neutralization assays. I believe this is stated in the Methods, but not Results.

Firstly, the abbreviation sVNT needs to be defined at its first appearance within the Results.

Secondly, there should be a more explicit statement of the authors' intent to bridge authentic virus assays to the surrogate assays.

Both of these goals could be accomplished in lines 153-156 as follows:

o "Live virus neutralisation strongly correlated with surrogate virus neutralisation assays (sVNT) in both plate ($r=0.89$) and bead formats ($r=0.9$) (Figure 1cd). These plate and bead sVNT assays were also well correlated for the ancestral virus ($r=0.85$, Figure 1e), VoC Beta ($r=0.83$) and Delta ($r=0.76$) (Supplementary figure 1ab).

- We have added the following section to the start of the results section:

"Plaque reduction neutralisation assays (PRNT) are a gold standard to assess antibody activity against blocking virus entry, however due to technical limitations the PRNT assay can be difficult to perform against a panel of viruses that have different replicative fitness and host range/cell lines. Furthermore, some sarbecoviruses have been sequenced but infectious virus not isolated (e.g. Bat CoV RaTG13) {Ge, 2016 #586}. We therefore used a surrogate virus neutralization test (sVNT) in a multiplex format, which uses recombinant protein of receptor binding domain (RBD) of different SARS family viruses to assess antibody inhibition of ACE2 binding.

And at the equivalent section (Line 153) describing correlations between PRNT and sVNT assays:

"Authentic SARS-CoV-2 virus based PRNT assays were assessed versus the plate and bead based sVNT assays, which have previously shown to account for more than 90% of total neutralising antibodies {Perera, 2021 #587; Premkumar, 2020 #589} across different immune cohorts {Chia, 2021 #588; Wang, 2022 #570}."

Reviewer #2 (Remarks to the Author):

Jia et al analysed cross-reactive neutralising responses against SARS-CoV-2 variants of concerns (VOC), SARS-CoV-2 related bat and pangolin viruses, SARS-CoV-1 and related bat sarbecoviruses. The authors analysed the magnitude and breadth of neutralising responses in sera from individuals vaccinated with two or three doses of BNT162b2 mRNA vaccine or inactivated SARS-CoV-2 vaccines (Coronavac or Sinopharm) with or without a history of previous SARS-CoV-2 or SARS-CoV-1 infection. They used a multiplex surrogate neutralisation assay based on receptor-binding domain. They found that hybrid immunity led to a high level of cross-reactive neutralising responses. High level of cross-reactive neutralising responses to VOC were also observed in individuals pre-infected with SARS-CoV-2 then vaccinated with BNT162b2. However, a previous infection with SARS-CoV-1 followed by BNT162b2 vaccination led to the highest magnitude of neutralising activity. Cross-reactive neutralising responses were also measured in individuals who experienced Omicron BA.2 breakthrough infection following vaccination with BNT162b2 vaccine and in individuals who received a third dose of BNT162b2 vaccine after 2 doses of the same vaccine or an inactivated vaccine.

This paper is relevant and well written. It is essential to determine how broad neutralising response can be elicited in order to lead a broad protection against future SARS-CoV-2 VOC and/or potential future pathogenic sarbecoviruses. The multiplex surrogate neutralisation assay looks appropriate and the limitations are well described. The strong point in this study is to have a few sera from individuals previously infected with SARS-CoV-1. The weakness of the paper is the figures. They are very difficult to understand because they are very busy, especially these ones showing the percentage of inhibition. The authors should show clear main figures to share their take home messages. Details can be added to supplementary figures.

Major comments:

- 1) It would great to include a supplementary figure showing the differences in RBD amino acids between SARS-CoV-2, SARS-CoV-1, bat and pangolin-related viruses studied in this paper. It would be a good reminder for the reader. It might help for the interpretation of the results.

We agree with this reviewer on the usefulness of a table comparing the different RBD conservation rates, and this has already been done for this panel in a related pre-print (Linfa Wang, 10.21203/rs.3.rs-1362541/v1), therefore we have added Table 2 as below, based on this data.

“The protein homology of the RBD panel (Table 2) {Wang, 2022 #570} ranges from 1 to 3 amino acid (aa) differences for a 99.6 to 98.7% conservation for VoCs Alpha to Mu, whilst Omicron BA.1 has 15 aa differences and is only 93.3% conserved versus SARS-CoV-2. Whilst clade 1 viruses have 55 to 60 aa differences in the RBD and are 73.1 to 75.3% conserved versus SARS-CoV-2.”

Table 2: RBD sequence homology versus ancestral SARS-CoV-2

SARS-CoV-2 RBD vs.	% aa conservation	Number aa difference
VoC Alpha	99.6	1
VoC Delta	99.1	2
VoC Beta	98.7	3
VoC Gamma	98.7	3
VoC Delta Plus	98.7	3
VoC Lambda	99.1	2
VoC Mu	98.7	3
VoC Omicron	93.3	15
Bat CoV RaTG13	90.1	22
Pangolin CoV Gx-P5L	86.6	30
SARS-CoV-1	73.1	60
Bat CoV WIV-1	75.3	55
Bat CoV RsSHC014	75.3	55
Bat CoV LYRa11	73.5	59
Bat CoV Rs2018B	74.9	56

- 2) The breadth of neutralising responses might also be impacted by the type of SARS-CoV-2 strain which infected people before vaccination. In this paper the authors focused on individuals pre-infected with the ancestral strain. Do the authors have access to samples from individuals pre-infected with VOC ? If so, it would be valuable to include these data in comparison.

No, there has been limited circulation of SARS-CoV-2 in Hong Kong, making it difficult to capture prior infection of variants in vaccinated individuals. Our infection prior to vaccination samples were collected in 2020, during the first year of the pandemic whilst variants of concern predominated from late 2020. The lineage genotype and infection date has been added to the supplementary subject information file. As there is a high aa similarity between the VoC (until Omicron) and ancestral SARS-CoV-2, and residual cross reactivity between VoC to ancestral viruses, we anticipate that there would be similar breadth of antibody responses in

infected+vaccinated subjects. For example, a Beta VoC infection primed individual with BNT162b2 vaccination compared to an Ancestral infection primed individual with BNT162b2 vaccination, due to the overwhelming titer of nAb that are generated. The vaccine type may outweigh the VoC prime, however this remains to be determined experimentally for various VoC.

This has been added to the final paragraph of the discussion point as below:

“Infection priming with different VoC (prior to Omicron) and subsequent Ancestral SARS-CoV-2 vaccination would likely generate similar breadth of pan-sarbecovirus responses due to residual cross-reactivity {Moyo-Gwete, 2021 #590;Moyo-Gwete, 2022 #591} and the overwhelming titer of nAb that are generated. The vaccine type may outweigh the VoC prime, however this remains to be determined experimentally for various VoC as imprinting does occur, it does not occlude new antibody responses.”

3) How did the authors exclude asymptomatic infections to SARS-CoV-2 which could impact on neutralising responses ? Did they measure anti-N ? A significant change in anti-N response between 2 timepoints could suggest asymptomatic infection. If so, it would be relevant to include these data in Supplementary data.

Hong Kong has had an intensive zero-COVID policy with only 12,000 cases in the first 2 years of the pandemic in a dense city of 7 million people. Intensive quarantine of cases and their contacts has meant asymptomatic infections have been identified by RT-PCR and isolated in quarantine. The seropositivity rate for SARS-CoV-2 was less than 1%, prior to January-April 2022 with a large Omicron BA.2 wave. N-based ELISA has been used on some donors as part of a larger sero surveillance study to screen for SARS-CoV-2 infection. Therefore, our donors infection status was RT-PCR and in some cases also N-ELISA confirmed. The N ELISA results where available and relevant are included in the supplementary data file.

This has now been stipulated in the methods and a paragraph added to the start of the discussion.

At discussion section:

“The antibody responses generated by diverse COVID-19 vaccines in naïve individuals or after infection with SARS-1 or VoC Omicron, boosting and heterologous vaccination provides an opportunity to address these challenges. Furthermore, Hong Kong has maintained a rigorous zero-COVID policy with low virus circulation until January 2022 (population based seroprevalence ~1%), and therefore in this study vaccine and infection immunogenicity was in a naïve population or with RT-PCR confirmed infection”

At methods section:

“To assess vaccine immunogenicity, plasma was collected from previously uninfected subjects (confirmed by N ELISA), prior to receiving the first vaccine dose and at 1 month post 2-dose vaccination with BNT162b2 or Coronavac (n=30).”

4) All figures showing the percentage of inhibition are too busy. I am wondering if the authors should only show the heat maps as main figures. Or they should split their panels showing the percentage of inhibition. In addition, sometimes 2 timepoints are shown in the same column of their histograms it is very confusing for the reader. Could the authors show one timepoint per column ? The slight difference in shape or color between each condition in the percentage of inhibition panels is also confusing. The authors should use a clear and comprehensive panel of colors and shapes.

The figures have been revised throughout for clarity. Reviewer 3 also raised this concern and apologies for our oversight. Pre/post 2 timepoint samples are now separated as two bars rather than overlaid together, and shaded bars given for post samples were relevant. The journal formatting in the reporting summary stipulates that individual data points should also

be shown. The study involves many different groups, and an exposure type is represented by a colour family (e.g., prior SARS-CoV-2 is purple, prior SARS-CoV-1 is orange)- enabling the final figure 5 data to be presented together. The text of Table 1 sample descriptions have been shaded for exposure type also.

An example of an updated figure for Figure 2C is shown below:

From:

To:

5) Could the authors mention the dosing interval between the 2 first doses of vaccine (mRNA or inactivated) ? The dosing interval may impact on neutralising responses so it is relevant to have this piece of information in methods.

Added to the methods section: "BNT162b2 vaccination is recommended 21 days apart and Coronavac 28 days apart".

Minor comment:

VoC or VOC: the authors should make sure they are consistent with abbreviations.

Updated VOC to VoC throughout.

Reviewer #3 (Remarks to the Author):

The manuscript by Jia, et al. describes the use a neutralization-surrogate assay to access the breadth of antibody responses following different sequences of exposure to coronavirus antigen. The study shows, in part, that BNT162b2 vaccination following prior exposure to SARS-CoV-1 led to antibody responses that were highly neutralizing against other sarbecoviruses including viruses of pandemic potential. Interestingly, this sequence appears to have resulted in better antibody responses against zoonotic strains than the Omicron BA.1 variant. The authors also show that in their hands, boosting with BNT162b2 improved antibody responses compared to boosting with CoronaVac. This was particularly true for individuals who received a two dose series of BNT162b2 as they did not notably increase their antibody response following vaccination with CoronaVac. The broadest and strongest antibody responses were seen in individuals vaccinated with either BNT162b2 or CoronaVac and then boosted with BNT162b2. While there are several novel findings in this manuscript and researchers and clinicians would find the information presented in this manuscript useful, several concerns should be addressed before publication. The methods section, and linked manuscripts, provide a clear path for anyone looking to repeat this work.

1. While well written, overall, the manuscript would benefit from additional editing. For example, line 84 beings, "It is about...", and I assume it should read, "It has about...". Additionally, line 110 should likely read, "...which led to an 11x increase...".

The manuscript has been further checked throughout for clarity and grammar. Highlighted text for updated grammar.

2. In general, and especially in figure two, the text on the figures and the size of the points on the graphs are sometimes too small to reliably resolve. This may just be an artifact of the format this reviewer used to view the figures.

Reviewer 2 also raised this concern, and apologies for oversight. We have reformatted figures for clarity, as bar graphs, to emphasise the mean/stdev and faded subject dots with shaded bars.

3. The correlation between the assays in figure 1 appears robust, the authors should perform (if not already done) and report the results of a Bland-Altman analysis to ensure that any systemic biases between the assays are accounted for.

A Bland-Altman analysis between sVNT for bead multiplex versus plate has been performed, as they are measured on the same scale. However, the Bland-Altman analysis may not be suitable for the correlation test between PRNT and sVNT data, because these two methods measure on different scales (PRNT takes values from 0-320, while sVNT takes values from 0-100) (discussed here, <https://stats.stackexchange.com/questions/167151/bland-altman-tukey-mean-difference-plot-for-differing-scales>). Whereas, the Spearman correlation may be

more appropriate for our data and can be used for both sVNT by both plate and bead multiplex and PRNT assays, it is a rank-based method and does not require data to be on the same scale.

As shown in the figure below, by Bland-Altman we found the average of the differences (between Plate sVNT and Bead sVNT) is 12.53 (95%CI: 9.39 - 15.68) units (blue area). This mean difference is not zero, and this means that on average the Plate sVNT measures 12.53 units more than the Bead sVNT. From the figure, we see that for samples with mean sVNT (x-axis) between 25 and 75, the bead sVNT generally had lower values than the Plate sVNT (difference, y-axis, higher than zero). It is possible to say that the bias is significant, because the line of equality (0 at y-axis) is not within the confidence interval of the mean difference (blue area).

The red and green areas show the limits of agreement, showing that we are convinced that 95% of the differences (between Plate sVNT and Bead sVNT) will be between -24.61 and 49.68:

Upper limit of agreement: 49.67594

Upper LOA- upper 95% CI: 55.06129

Upper LOA- lower 95% CI: 44.2906

Lower limit of agreement: -24.6082

Lower LOA- upper 95% CI: -19.22285

Lower LOA- lower 95% CI: -29.99354

Generally, from the Bland-Altman analysis, we found Bead sVNT had lower values compared to Plate sVNT. This bias does not seem to be systematic, instead may suggest biological differences between the two assays, as the bias was not consistently positive or negative within the whole range of x-axis. There are technical differences between these assays, with a 1:10 dilution used for plate sVNT and 1:20 dilution for multiplex sVNT, and protein conformation in a 3D form versus static plate bound, which may account for these differences. Judging from the plot, the bias is non-linear and can not be adjusted by simple linear functions. One should note that although the biases between two assays are non-zero and non-linear, the Spearman correlation (rank-based) is significant between them (Figure 1E), suggesting the agreement between two methods by rank are good.

The interpretation of the results is mainly based on this reference

(<https://www.ncbi.nlm.nih.gov/pmc/articles/PMC4470095/>).

Figure. The Bland Altman analysis with the representation of 95% confidence interval limits for mean and agreement limits (colored areas). The analysis was performed using package "blandr" in R for plate and bead sVNT results from Figure 1E.

This figure above has been added as Figure 1F, and described in the results:

“However, Bland Altman analysis for assay comparability between the plate format and bead format sVNT assays (Figure 1f), showed that the plate sVNT measures 12.53 units more than the bead sVNT. Although the biases between two assays are non-zero and non-linear by Bland Altman, the Spearman correlation (rank-based) is significant between them (Figure 1E), suggesting the agreement between two methods by rank are good.

4. While understandable given the limitations of the various assays, it remains concerning that a different assay platform was used for the Omicron BA.1 results. This should be discussed more clearly in the limitations section.

We have added the below sentence to the limitations discussion of the sVNT assay

“sVNT assays are variant dependent, and Omicron RBD antibodies were assessed by both bead and plate approaches. Results are shown for the plate-based method due to better correlation with gold standard PRNT results. All other RBDs were represented in the multiplex assay simultaneously.”

5. In the limitations section, the authors should also point out that their primary assays only look at spike and therefore ignore any potential neutralization/protective antibodies against other virus proteins.

The limitation sections a) and b) refer to the limitations of studying RBD-antibodies only.

Section a) describes that immune functions of non-RBD antibodies “does not assess neutralizing activity directed to other known regions of the S protein, including the N terminal domain (NTD), the S2 domain, or S-2P-ecto domains”

The section b) described that “T cell responses⁸ and non-neutralizing and their effector functions^{9,10} also contribute to protection against severe disease”, and has been modified to include “but are not covered in our study.”

REVIEWERS' COMMENTS

Reviewer #1 (Remarks to the Author):

This reviewer greatly appreciates the efforts the authors have undertaken to address the concerns of all reviewers. This manuscript is an excellent addition to the field. I have no further concerns to address.

Reviewer #2 (Remarks to the Author):

The authors perfectly replied to my comments. The figures are clear now.

Reviewer #3 (Remarks to the Author):

The revised manuscript by Jia, et al. does an impressive job of addressing the recommendations made by myself and the other reviewers. The manuscript describes the use of a neutralization surrogate assay to assess the breadth of antibody responses following exposure to different sequences of coronavirus antigen. Of note, the authors demonstrate that previous exposure to SARS-CoV-1 leads to antibody responses that are highly neutralizing against sarbecoviruses of pandemic potential. The revised manuscript clarifies several points of concern made the reviewers. The authors included numerous additional analyses that strengthen the manuscript and provide an even greater context for the significance of their findings.

I have no additional recommendations.